# Influence of Anthropogenic Factors on the Diversity and Structure of a Dry Forest in the Central Part of the Tumbesian Region (Ecuador–Perú)

**Jorge Cueva Ortiz [1,\*]**, **Carlos Iván Espinosa [2]**, **Carlos Quiroz Dahik [1]**,
**Zhofre Aguirre Mendoza [3]**, **Eduardo Cueva Ortiz [2]**, **Elizabeth Gusmán [2]**, **Michael Weber [1]**
and **Patrick Hildebrandt [1]**

1   Institute of Silviculture, TUM School of Life Sciences Weihenstephan, Technische Universität München,
    85354 Freising, Germany; caquirozd@hotmail.com (C.Q.D.); michael.weber@wzw.tum.de (M.W.);
    hildebrandt@tum.de (P.H.)
2   EcoSs_Lab, Departamento de Ciencias Biológicas, Universidad Técnica Particular de Loja, San Cayetano
    Alto, 110107 Loja, Ecuador; ciespinosa@utpl.edu.ec (C.I.E.); ecuevafcsf@hotmail.com (E.C.O.);
    ecguzman@utpl.edu.ec (E.G.)
3   Carrera de Ingeniería Forestal, Universidad Nacional de Loja, Ciudadela Guillermo Falconí E.,
    110-110 Loja, Ecuador; zhofre.aguirre@unl.edu.ec
\*   Correspondence: jorge.cueva@tum.de; Tel.: +49-8161-71-4690

**Abstract:** The dry forest of southern Ecuador and northern Perú (called the Tumbesian region) is known for its high diversity, endemism, and healthy conservation state. Nevertheless, the forest is exposed to many threats linked with human activities. To understand the effects of these threats, which have not been appropriately assessed, we pose two questions: (a) What are the diversity and structural situations of the forest? (b) Are anthropogenic activities affecting the composition and structure of the forest? The assessed factors were species richness, diversity, species similarity, abundance, and density. Forest information was obtained from 72 plots (total area 25.92 ha) randomly placed to cover a wide range of stand densities (from 200 to 1100 m a.s.l.). After constructing linear mixed models and selecting the most influential one, we determined the individual influences of 12 predictors. The human pressure index (HPI) was the most negative predictor of forest health, and annual precipitation was the most important abiotic predictor of good health conditions. Livestock grazing did not significantly change the diversity and structure of mature forest. The mean annual temperature and stoniness influenced only the basal area and number of individuals, respectively. The species composition in our study area was not affected by the HPI, but was strongly predicted by annual precipitation.

**Keywords:** Ecuador; Perú; dry forest; Tumbesian region; species richness; diversity; similarity; goats; human pressure

## 1. Introduction

In a global assessment of predominant habitat types, Olson [1] identified 14 major terrestrial habitat types. Among these, tropical and subtropical dry forest ecosystems collectively represent 42% of all tropical forests [2].

The remains of tropical dry forests presently cover 1,048,700 km$^2$. Most of these remains (54.2%) are found in South America, mainly in northeastern Brazil, in the southeast of Bolivia and Paraguay, and north of Argentina. These regions are the largest continuous dry forests registered on the globe [2]. Other important but smaller areas are found in the Yucatan peninsula (Mexico), in the northern parts of

Venezuela and Colombia, and in Asia (Thailand, Vietnam, Laos, and Cambodia). Tropical dry forests with smaller coverage have also been reported in the north of Panama, the west of Mexico, Australia, Africa, and along the Pacific coast of South America, between Ecuador and Perú [3–5]. Isolated areas exist in the inter-Andean valleys of Ecuador, Perú, Colombia, and Bolivia.

Tropical dry forests are well-known as important ecosystems, yet they are exposed to many threats [1–3,6–9], including deforestation, fragmentation, overgrazing, fire, conversion to agriculture, and drought. These factors are mainly responsible for loss of species (or reduced genetic variability); however, they can also increase the abundance of other species, thereby homogenizing the landscape and the ecosystem [10–12]. A clear instance of these effects was shown by Semper et al. [8], who revealed the influence of land conversion on the richness of animals in Chaco, Argentina: the local extinction of birds and mammals here was estimated as 56% in areas recently turned to pastures, and 29% in areas converted to crop fields.

The future of dry forests has roused great concerns. This is because dry forests are highly accessible owing to their convenient topographic and climatic conditions, making them easy targets for homogenization, which reduces the genetic variability both within species and among populations [12].

The Tumbesian region is a substantial area (approximately 135,000 km$^2$) of dry forest located on the coast of the southwestern part of Ecuador, which extends to the northwest of Perú [5]. The Tumbesian region belongs to the "Chocó/Darien Western Ecuador", an area identified as one of the 25 global biodiversity hotspots with a large number of endemic species (85 birds, 60 mammals, 63 reptiles, 210 amphibians, and 2250 plants) [13]. For this reason, Olson [1] recommended including this area in the regional strategy plans for the "Tumbesian–Andean Valleys Dry Forests" in the Terrestrial Ecoregion.

Although dry forests have been characterized in several studies, their functionality remains poorly understood [14]. This applies to equatorial forests in general, but mainly in the Ecuadorian–Peruvian dry forest [15]. In particular, there exists a lack of knowledge regarding the effects of human activities such as cattle raising or selective timber extraction on the diversity and structure of the Ecuadorian dry forest.

In the Ecuadorian dry forests, seven distinct formations has been identified [5,16]—spiny dry scrub, deciduous dry forest, semi-deciduous dry forest, low montane dry forest, southern inter-Andean dry forest, eastern inter-Andean dry forest, and northern inter-Andean dry forest—of which semi-deciduous dry forest (located between 200 and 1100 m a.s.l.) is the most diverse formation, followed by deciduous dry forest (located from sea level to 700 m a.s.l.). For this reason, our study was carried out in these two formations.

Currently, local livestock farmers and conservationists from the study region are in the midst of an intense discussion. The first group is arguing that livestock grazing supports conservation, because animal sales generate income for families and hence prevent forest clearance for other purposes [17]; furthermore, grazing reduces the fuel load and risk of forest fires, as reported in Central America's dry forest [3,18]. In contrast, some conservationists argue that extensive grazing impedes the natural regeneration of forests, affecting their structure and diversity [19]. Owing to the lack of suitable information, no political recommendations for livestock handling are available.

Our study strives to improve the current knowledge on dry forests and to identify whether human activities significantly impact forest conditions. To this end, we pose the following research questions: (a) What are the structural and diversity situations of the dry forest in the central zone of the Tumbesian region? (b) Are anthropogenic activities affecting the diversity and structure of the tropical dry forest in this region?

## 2. Materials and Methods

*2.1. Study Area*

The research was conducted in the dry forest of the cantons Macará and Zapotillo of the Loja province in southern Ecuador, and the department and province of Tumbes in the northwest of Perú. This area constitutes the central part of the Tumbesian region (Figure 1). The altitudinal range of the deciduous and semi-deciduous dry forest formations varies between 200 and 1100 m a.s.l. [5,7]. The annual mean temperature in the area ranges between 20 °C and 26 °C, and the annual rainfall ranges from 300 mm in the lower lands to 700 mm in the highlands [19].

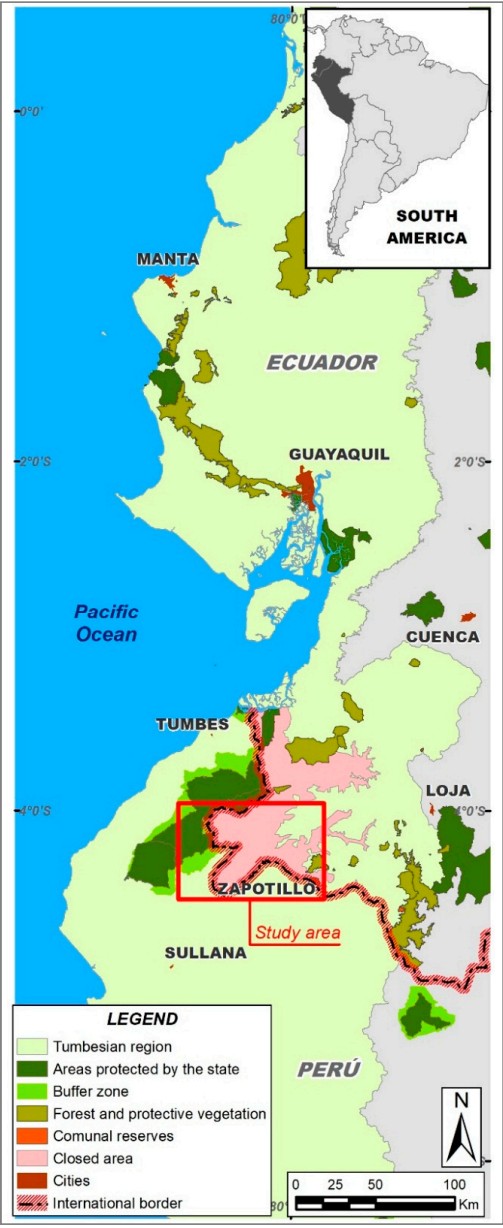

**Figure 1.** Study area in the Tumbesian region. Modified from "*La Región Tumbesina–una riqueza compartida*" [20].

Some species characteristic to the study area have been reported: *Ceiba trichistandra* (A.Gray) Bakh., *Cavanillesia platanifolia* (Bonpl.) Kunth, *Eriotheca ruizi* (K. Schum.) A. Robyns, *Handroanthus chrysanthus* (Jacq.) S.O.Grose, *Terminalia valverdae* A.H. Gentry, *Bursera graveolens* (Kunth) Triana & Planch.,

and *Piscidia carthagenensis* Jacq., among others [14,21]. Likewise, some species endemic to Ecuador and Perú have been recorded: *Ficus jacobii* Vázq. Avila, *Coccoloba ruiziana* Lindau, *Mauria membranifolia* Barfod & Holm-Niels., *Armatocereus brevispinus* Madsen, and others [21]. Nevertheless, several of them have been exposed to strong extraction processes.

During the 60s and 70s, many *H. chrysanthus* and *Handroanthus billbergii* (Bureau & K.Schum.) S.O.Grose were extracted for supplying the parquet industry. Accordingly, the coastal forests of southwestern Ecuador up to 1000 m a.s.l. were declared "closed areas" in May 1978 [22]. Although logging continues to operate today, the declaration has substantially decreased this activity. During the last few decades, the natural cover in the study area has dramatically changed: deforestation rates of 33% for seasonally dry forest and 18% for shrublands were reported between 1978 and 2008. In both cases, the area has been mainly converted into pastures [23]. Another relevant problem in the region is the increasing clearing of new areas for crop plantations [23]. The Zapotillo irrigation system, which came into operation approximately 8 years ago, has stimulated the cultivation of new agricultural fields and increased the number of free-grazing goat and bovine herds in nearby forests [24]. Finally, in recent years, the province has experienced extreme climatic conditions (drought or water saturation due to the El Niño phenomenon) that will probably have future impacts [25] on diversity and species composition, the recurrence of which would enhance their effects.

## 2.2. Data Collection

### 2.2.1. Forest Inventory

Following the information provided by Aguirre et al. [5], Lozano [16], and Cueva et al. [26], we separated the study area into strata [27]. In our inventory we considered two formations (deciduous and semi-deciduous) and three density levels (dense, semi-dense, and sparse) in each type. This resulted in six different strata, which were considered to equivalently cover most of the forest variability.

In every stratum, four inventory clusters comprising three plots of 60 m × 60 m (each with a nested subplot of 20 m × 20 m) were arranged in an L-shape (Figure 2) and randomly distributed, as described in the methodology of the National Forestry Assessment of Ecuador [28]. However, for more detailed information, we reduced the diameter at breast height (DBH) of the tree inventory from 20 cm to ≥10 cm. Within the nested subplots, we also registered trees between 5 and 10 cm DBH and shrubs with ≥5 cm DBH. In addition, we collected and weighed livestock excrement within a 1 m × 60 m transect along the eastern edge of each plot as an indicator of grazing intensity. Thus, we inventoried 24 clusters (20 in Ecuador and four in Perú) containing 72 plots, which altogether covered 25.92 ha.

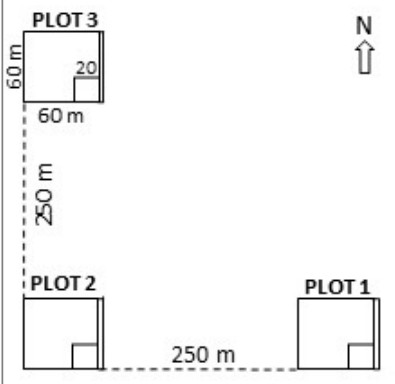

**Figure 2.** Cluster design. Shrubs were inventoried in the subplot in the southeast corner of each plot, and animal excrement was collected from the transect at the eastern edge.

In each plot, all trees and shrubs were labeled and their species, DBH, and total height were recorded. The species were identified in the field, and where necessary, botanical samples were collected for subsequent identification in the herbarium of the Universidad Técnica Particular de Loja (UTPL). The scientific names were reviewed and corrected based on Reference [29].

### 2.2.2. Biotic and Abiotic Factors

The number of grazing animals in the forest was estimated on the basis of the excrement amounts. Goats and cattle feces were separated, as these animals were the most common livestock in the study area. The excrement of horses and donkeys (Equine) was also collected, but was summed for each plot. Cattle, horses and donkeys are common throughout the area, but it is difficult to find goats in the highest zones.

As a proxy of the intensity of anthropogenic impact on the forest (due to activities such as woody extraction, grazing, and trampling), we computed the human pressure index (HPI). Adapted from Hegyi's competition index [30], the HPI is defined as:

$$\text{HPI} = \sum_{i=1}^{n} \frac{F_i}{B \times dist_i}. \tag{1}$$

where:

$F_i$ = number of families in the *ith* neighboring village;
$B$ = total basal area of trees and shrubs per plot, projected to hectares (m$^2$/ha);
$dist_i$ = horizontal distance from the *ith* neighboring village to the studied plot (m).

To calculate the HPI, we considered all neighboring villages around each plot within a distance of 3 km, and collected information regarding the number of families in each village. The horizontal distances between the neighboring villages and the respective plots were calculated on a QGIS Desktop 2.8.7 [31]. The basal areas of trees and shrubs in our inventory data were used to consider the competitiveness among the vegetation at each site. The anthropogenic pressure ranged between 0 (no pressure) and 0.01367. To avoid a circular argument, the basal area was excluded from the HPI when assessing the structural aspects (i.e., number of individuals and basal area). In this case, the pressure ranged between 0 and 0.04117.

Herein, climatic data reporting gridded climate data to 30-arc second resolution (approximately 1 km$^2$) were used [32]. At each plot, we recorded the total annual precipitation (Ann.Prec), precipitation in the wettest month (Mth.Prec), and the mean annual temperature (Temper). In the study area, the ranges of Ann.Prec, Mth.Prec, and Temper were 442–1271 mm, 180–312 mm, and 22.39 °C–24.63 °C, respectively.

Ecuadorian part soil information was obtained from the Geopedological Map of Zapotillo and Celica of the Instituto Espacial Ecuatoriano [33]. The soil parameters were soil depth (five categories: 0–10, 11–20, 21–50, 51–100, and >100 cm), drainage (three categories: poor, moderate, and good), stoniness (five categories: without, very few, few, frequent, and abundant), and texture (four categories: loam, sandy loam, clay loam, and clay–sandy loam). The Peruvian part soil type was identified from the Soil Classification Map of the Oficina Nacional de Evaluación de Recursos Naturales (ONERN) [34]. As specific soil characteristics were unavailable, they were assumed to be identical to those in the nearest area with the same soil type on the Ecuadorian side.

Values of each predictor can be found as Supplementary Material (Table S1).

### 2.3. Data Analysis

### 2.3.1. Data Processing

Species richness was calculated as the total number of species of trees and shrubs in each plot. Diversity was calculated using the Simpson reciprocal index, which is considered to be one of most

effective and robust diversity measures [35]. Furthermore, the species accumulation curve was obtained by calling the function specaccum() in the R programming environment [36] using the Coleman method [35] in the "vegan" package [37].

To correct for the different plot sizes, the numbers of individuals were projected onto numbers of individuals per hectare before adding the numbers of trees and shrubs. Individuals' distribution by diametric classes were obtained for the forest and for some species, because they could be indicators of anthropogenic activities. Basal area of a tree was computed using the equation $0.7854 \times DBH^2$ [38]; the total basal area was obtained by summing the basal areas of all individual trees and shrubs inside the plot and projected to ha. Multi-stemmed individuals whose branches had minimum DBH were treated as the same individual to compute most parameters, whereas to compute the basal area per plot, each stem with minimum DBH was treated as a separate individual.

The species similarity among plots of the same cluster was determined by computing the Sørensen index [35] in the "vegan" package [37]. The Sørensen index tells us how similar two samples are, in terms of presence/absence of species [35].

2.3.2. Effects on Diversity, Structure, and Species Similarity

To determine whether the diversity, structure, and species similarity are affected by anthropogenic activities, climate, and/or soil characteristics in the study area, we applied generalized linear mixed models (GLMMs) or linear mixed models (LMMs) [39–42]. We used species richness and the Simpson reciprocal index as diversity indicators. When identifying the effect of anthropogenic activities on similarity, we used the Sørensen index as the dependent variable. In addition, the number of individuals and basal area per ha were the response variables when analyzing impacts on the forest structure.

In all cases, we considered 12 predictor variables: four biotic indicators (Goats, Cattle, Equine, HPI), four abiotic factors (Temper, Altitude, Ann.Prec, Mth.Prec), and four soil variables (Soil.depth, Drainage, Stoniness, Texture). Pearson's correlation coefficient was calculated for all combinations of variables (Table 1). To avoid collinearity in model building, strongly correlated variables were not included in the same model. Furthermore, overfitting was avoided by limiting the number of variables in any model to five [43] (Table A1). Formation and cluster were considered as random effects, taking into account that plots are nested within clusters, whereas clusters are nested within formations. Interactions between variables were not considered because we wished to identify variables' influence per se, the purpose was to facilitate the implementation of mitigation measures in the field, if necessary.

The influence of each predictor on species richness was estimated in GLMMs with the Poisson error distribution and Laplace approximation. Laplace approximation was used to achieve true likelihood, which cannot be obtained through penalized quasi-likelihood (PQL) approach [40,44]. The influence of the predictor variables on species similarity, the Simpson index, number of individuals per hectare, and basal area was assessed via the restricted maximum likelihood approach (REML), which was chosen because we were dealing with continuous response variables and normal distribution [40].

The best models were chosen using the delta Akaike information criterion ($\Delta AIC \leq 2$) [45], and the residuals distribution of each best model are presented as Supplementary Material (Figures S1–S5). We also computed the marginal and conditional variances (*R2m* and *R2c*, respectively) [46] using the MuMIn package [47] in R, which provides the proportion of the variance explained by the models. The models were tested for overdispersion.

To improve the model fittings, the abiotic variables and altitude were log-transformed and the HPI was transformed using the square root function. For comparison, the excrement amount of each type of livestock was standardized to 1.

All GLMMs and LMMs analyses were performed using the lme4 v 1.1-14 package [39] in the R v 3.4.0 programming environment [36].

**Table 1.** *p*-Values obtained in the Pearson's correlation test of predictor variables. Correlated variables are shaded.

| | Goats | Cattle | Equine | HPI | Temper | Altitude | Ann.Prec | Mth.Prec | Soil.depth | Drainage | Stoniness | Texture |
|---|---|---|---|---|---|---|---|---|---|---|---|---|
| **Goats** | 0 | | | | | | | | | | | |
| **Cattle** | 0.266 | 0 | | | | | | | | | | |
| **Equine** | 0.367 | 0.338 | 0 | | | | | | | | | |
| **HPI** | 0.464 | 0.044 | 0.794 | 0 | | | | | | | | |
| **Temper** | 0.120 | $5.2 \times 10^{-7}$ | 0.004 | $2.1 \times 10^{-4}$ | 0 | | | | | | | |
| **Altitude** | 0.107 | $7.2 \times 10^{-5}$ | 0.021 | 0.023 | $1.0 \times 10^{-28}$ | 0 | | | | | | |
| **Ann.Prec** | 0.111 | 0.160 | 0.108 | 0.816 | $1.3 \times 10^{-4}$ | $3.1 \times 10^{-11}$ | 0 | | | | | |
| **Mth.Prec** | 0.701 | 0.381 | 0.635 | 0.010 | 0.355 | $6.4 \times 10^{-5}$ | $1.7 \times 10^{-14}$ | 0 | | | | |
| **Soil.depth** | 0.578 | 0.012 | 0.181 | 0.000 | 0.000 | 0.000 | 0.034 | 0.136 | 0 | | | |
| **Drainage** | 0.412 | 0.000 | 0.011 | 0.023 | 1.0E-03 | 0.020 | 0.551 | 0.397 | 0.191 | 0 | | |
| **Stoniness** | 0.912 | 0.536 | 0.831 | 0.949 | 0.000 | 0.000 | 0.000 | 0.000 | 0.000 | $1.0 \times 10^{-3}$ | 0 | |
| **Texture** | 0.795 | 0.005 | 0.159 | 0.000 | 0.000 | 0.000 | 0.000 | 0.000 | 0.000 | 0.040 | 0.013 | 0 |

Several ways to assess the predictors' influence on one response variable are available, for instance, the stepwise procedure has been frequently used, however, it has been bypassed due to a lack of confidence in the model selection [45,48]. Without intending to criticize the procedure, we preferred to use a widely reliable process [40,41].

## 3. Results

### 3.1. Tree and Shrub Species Diversity, Composition Similarity, and Forest Structure

We recorded 7815 individuals from 91 species belonging to 34 families (Table A2). From these species, 71 were identified up to species level, 11 up to genus level, and 3 up to family level only. Six species (totaling 49 individuals) were unidentifiable and were classified as unknown. Leguminosae was the most diverse family with 24 species, followed by Malvaceae with six species. The species with the highest number of individuals were *Handroanthus chrysanthus* (1662) and *Piscidia carthagenensis* (1042).

The examined plots contained 4–27 species (Figure 3a). In the species accumulation curve (Figure 3b), 67% of the species were registered in 15 plots, and 90% were found in 45 plots. Similar results were presented in Aguirre [49]. The Simpson reciprocal indices ranged from 1.88 to 11.59 (Figure 3c). These results confirm large species diversity in the forests, reflecting the different species compositions and density landscapes in the study area.

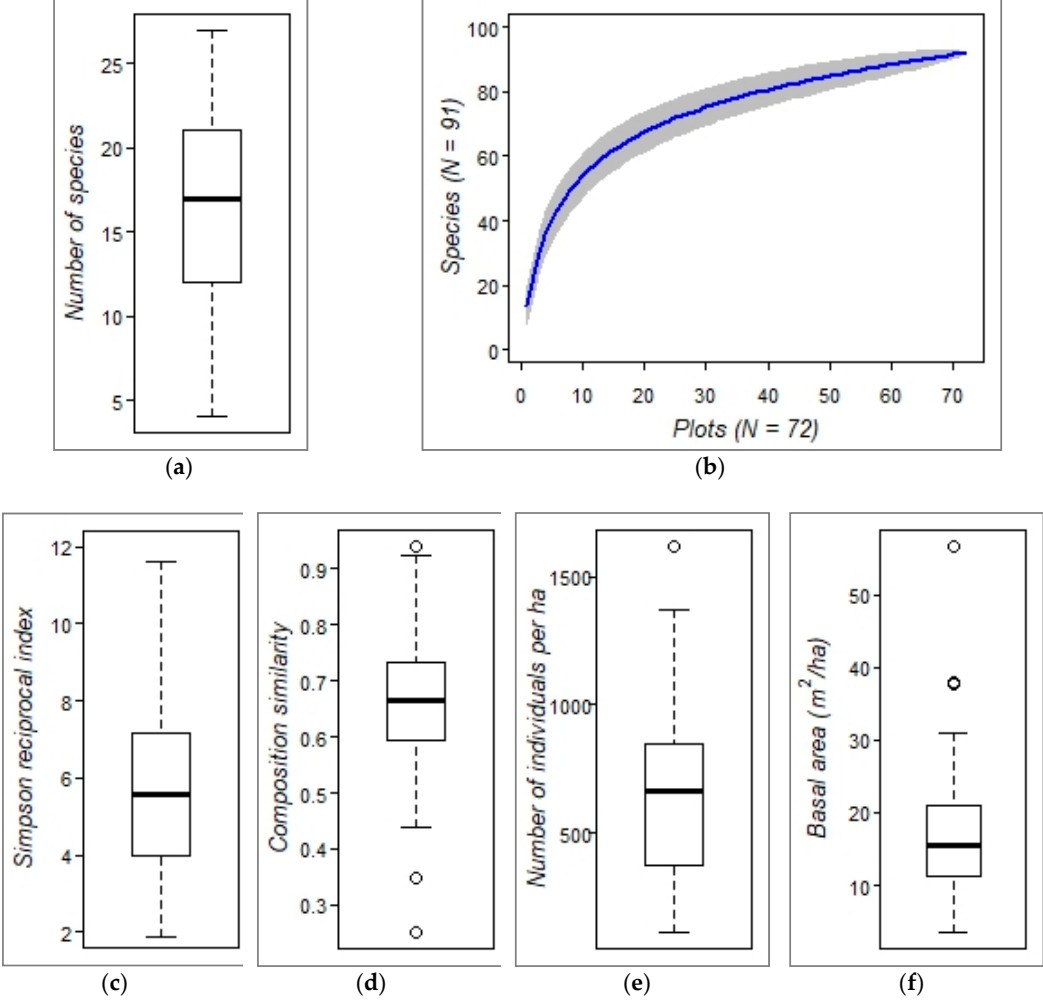

**Figure 3.** Current diversity and structure situations of the forests in the study area: (**a**, **c–f**) ranges of species parameters in the plots; (**b**) species accumulation curve obtained via the Coleman method with confidence of 95% (grey buffer).

The Sørensen indices of the clusters (>0.6 in most clusters; see Figure 3d) evidence high similarity among the plots in each cluster. This result can be explained by the close proximity of the plots. In the single exception with low similarity among its plots, one plot was separated by a brook and located on steep land, and was hence difficult to access.

Overall, the structural parameters of the forest were found to be quite diverse. The average DBH ranged from 15.2 to 33 cm. The number of individuals per hectare ranged between 108 individuals in plots located in sparse forests (the lowest and driest parts of the ecosystem) to 1620 individuals per hectare in forest regions with higher vegetal cover (Figure 3e). The basal area ranged from 3.5 to 56.7 m² per hectare (Figure 3f).

The diameter frequency distribution of all individuals followed a negative exponential relationship, as expected in natural forests (Figure 4a). However, some species distinctly deviated from this trend; for instance, the DBH exceeded 80 cm in many *Ceiba trichistandra* individuals (Figure 4b), was typically in the mid-range (15–55 cm) for *Cavanillesia platanifolia*, *Bursera graveolens*, and *Loxopterygium huasango* Spruce ex Engl. (Figure 4c–e), and never exceeded 45 cm in *H. chrysanthus* and *P. carthagenensis* (Figure 4f,g).

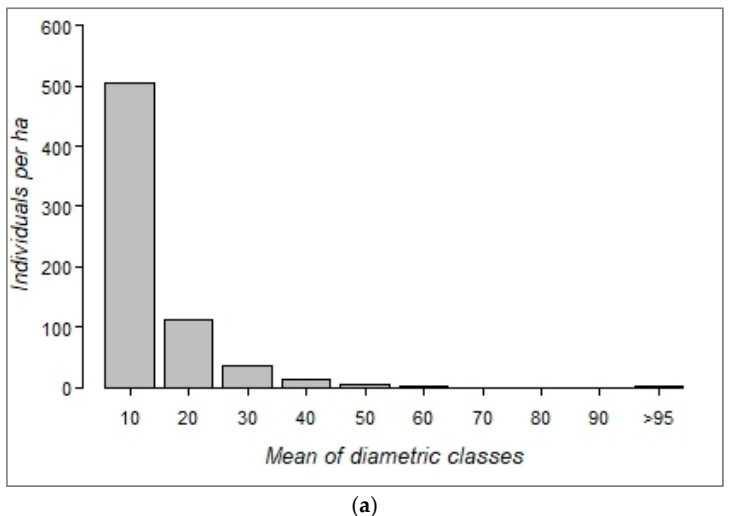

(**a**)

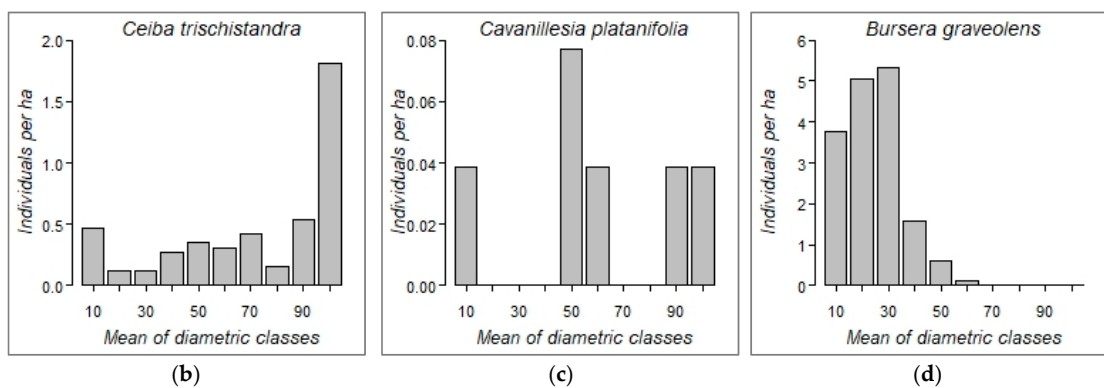

(**b**)　　　　　　　　　　　　(**c**)　　　　　　　　　　　　(**d**)

**Figure 4.** *Cont.*

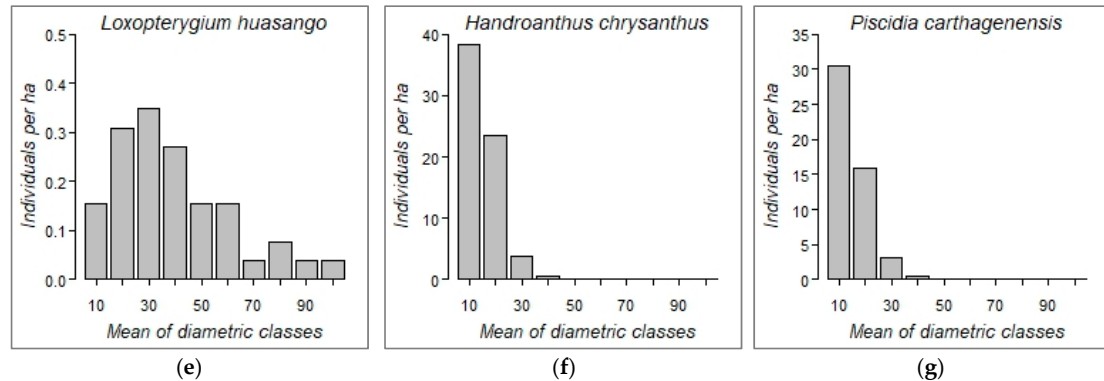

**Figure 4.** Diameter class distributions of individuals per ha: (**a**) all species; (**b**–**g**) distributions of selected species.

To briefly compare the formations: we recorded 4012 individuals from 65 species, with a diversity index that varied from 2.86 to 9.81 for deciduous forest, and 4833 individuals from 82 species and Simpson reciprocal indices from 1.88 to 11.59 in semi-deciduous forest. 25 families were recorded in the deciduous formation and 34 in the semi-deciduous formation. The basal area ranged from 3.70 to 37.71 m$^2$/ha in deciduous and from 345 to 56.72 m$^2$/ha in semi-deciduous forest.

### 3.2. Anthropogenic Influences on Species Diversity, Similarity, and Forest Structure

Table 2 shows the best models for each response variable together with all models with ΔAICs below 2. The anthropogenic variables and annual precipitation strongly influenced all analyzed parameters.

**Table 2.** Best models of the five response variables. df = degrees of freedom; AIC = Akaike's information criterion; ΔAIC = Delta Akaike's information criterion; *R2m* = marginal variance explained by fixed effects; *R2c* = conditional variance explained by random and fixed effects. In models with HPI.BA, the basal area of trees and shrubs was included in the human pressure index.

| Model | df | AIC | ΔAIC | *R2m* | *R2c* |
|---|---|---|---|---|---|
| **Richness** | | | | | |
| ~1 + HPI.BA + Ann.Prec | 4 | 412.44 | 0 | 0.26 | 0.63 |
| ~1 + HPI.BA | 3 | 413.38 | 0.95 | 0.21 | 0.63 |
| ~1 + Goats + HPI.BA + Ann.Prec | 5 | 414.33 | 1.89 | 0.26 | 0.63 |
| ~1 + Equine + HPI.BA + Ann.Prec | 5 | 414.34 | 1.90 | 0.26 | 0.63 |
| ~1 + Cattle + HPI.BA + Ann.Prec | 5 | 414.38 | 1.94 | 0.26 | 0.63 |
| ~1 | 2 | 424.54 | 12.10 | 0 | 0.56 |
| **Diversity** | | | | | |
| ~1 + Goats + Cattle + Equine + HPI.BA + Ann.Prec | 8 | 282.50 | 0 | 0.28 | 0.31 |
| ~1 + Goats + Ann.Prec + Drainage | 7 | 282.66 | 0.17 | 0.31 | 0.37 |
| ~1 + Goats + Cattle + HPI.BA + Ann.Prec | 7 | 282.67 | 0.17 | 0.27 | 0.32 |
| ~1 + Goats + Equine + HPI.BA + Ann.Prec | 7 | 283.61 | 1.11 | 0.31 | 0.31 |
| ~1 + Goats + HPI.BA + Ann.Prec | 6 | 283.81 | 1.31 | 0.30 | 0.31 |
| ~1 + Cattle + Equine + HPI.BA + Ann.Prec | 7 | 284.48 | 1.99 | 0.27 | 0.31 |
| ~1 | 3 | 311.06 | 28.56 | 0 | 0.20 |
| **Species similarity** | | | | | |
| ~1 + HPI.BA + Ann.Prec | 5 | −69.54 | 0 | 0.17 | 0.29 |
| ~1 + HPI.BA | 4 | −68.74 | 0.80 | 0.10 | 0.27 |
| ~1 + Temper + Mth.Prec | 5 | −68.19 | 1.35 | 0.08 | 0.31 |
| ~1 + Ann.Prec | 4 | −67.88 | 1.66 | 0.11 | 0.30 |
| ~1 | 3 | −66.30 | 3.24 | 0 | 0.28 |

**Table 2.** *Cont.*

| Model | df | AIC | ΔAIC | R2m | R2c |
|---|---|---|---|---|---|
| **Number of individuals** | | | | | |
| ~1 + Goats + Cattle + HPI + Stoniness | 7 | 959.85 | 0 | 0.20 | 0.54 |
| ~1 + Goats + Equine + HPI + Stoniness | 7 | 960.57 | 0.72 | 0.20 | 0.54 |
| ~1 + Cattle + Equine + HPI + Stoniness | 7 | 961.42 | 1.58 | 0.19 | 0.54 |
| ~1 | 3 | 1010.08 | 84.38 | 0 | 0.56 |
| **Basal area** | | | | | |
| ~1 + Temper + Mth.Prec | 5 | −25.59 | 0 | 0.17 | 0.36 |
| ~1 + HPI + Ann.Prec | 5 | −25.16 | 0.43 | 0.23 | 0.33 |
| ~1 + Temper | 4 | −24.71 | 0.89 | 0.12 | 0.35 |
| ~1 + Equine + HPI + Ann.Prec | 6 | −23.60 | 1.99 | 0.26 | 0.35 |
| ~1 | 3 | −17.64 | 7.95 | 0 | 0.33 |

In the goodness of fit tests for species richness, 21%–26% of the variances were explained by fixed predictors (*R2m* in Table 2), and up to 63% of the variances were explained by both random and fixed predictors (*R2c* in Table 2). In the models of the Simpson index, 27%–31% of the variances were explained by fixed predictors and 30%–36% of the variances were attributed to fixed and random predictors. In the structural parameters, fixed and fixed + random predictors explained 19%–20% and 54% of the variances in terms of number of individuals respectively; in terms of basal area, fixed and fixed + random predictors explained approximately 26% and 35% of the variances, respectively. Overall, these results confirm a high contribution of fixed and random factors to the variations in the richness and number of individuals, but a lesser contribution to the variations in the Simpson index, species similarity and basal area.

Species richness is highly influenced by the HPI, annual precipitation, and presence of animals (Table 2). In the five models with ΔAIC below 2, the HPI was presented in all models and annual precipitation was included in four models. Both factors were predictors of the best model: the HPI exerted a significant negative influence (Table 3, Figure 5a). The annual precipitation exerted a positive but insignificant influence on species richness (Table 3), which seemed to culminate in the highest species number at an intermediate level of precipitation (Figure 5b).

**Table 3.** Influence of predictors included in the best models on richness, diversity, species similarity, number of individuals, and basal area according to GLMM, *n* = 72.

| Predictor | Estimates | Std. Error | p (<0.05) | |
|---|---|---|---|---|
| **Richness** | | | | |
| HPI.BA | −7.82 | 2.22 | $2.4 \times 10^{-4}$ | *** |
| Ann.Prec | 0.77 | 0.44 | 0.09 | † |
| **Diversity** | | | | |
| Goats | 1.78 | 1.83 | 0.33 | |
| Cattle | −1.28 | 1.17 | 0.40 | |
| Equine | 0.55 | 1.03 | 0.49 | |
| HPI.BA | −20.38 | 9.34 | 0.03 | * |
| Ann.Prec | 7.74 | 1.72 | $1.2 \times 10^{-5}$ | *** |
| **Species similarity** | | | | |
| HPI.BA | −0.31 | 0.13 | 0.02 | * |
| Ann.Prec | −0.64 | 0.30 | 0.03 | * |
| **Number of individuals** | | | | |
| Goats | −205.80 | 266.51 | 0.42 | |
| Cattle | −143.93 | 162.62 | 0.38 | |
| HPI | −2411.17 | 777.20 | $2.9 \times 10^{-3}$ | ** |
| Stoniness | −20.67 | 32.84 | 0.50 | |

**Table 3.** *Cont.*

| Predictor | Estimates | Std. Error | *p* (<0.05) | |
|---|---|---|---|---|
| **Basal area** | | | | |
| Temper | −4.92 | 2.06 | 0.02 | * |
| Mth.Prec | 0.67 | 0.39 | 0.07 | † |

*** $p < 0.001$; ** $p < 0.01$; * $p < 0.05$; † $p < 0.1$.

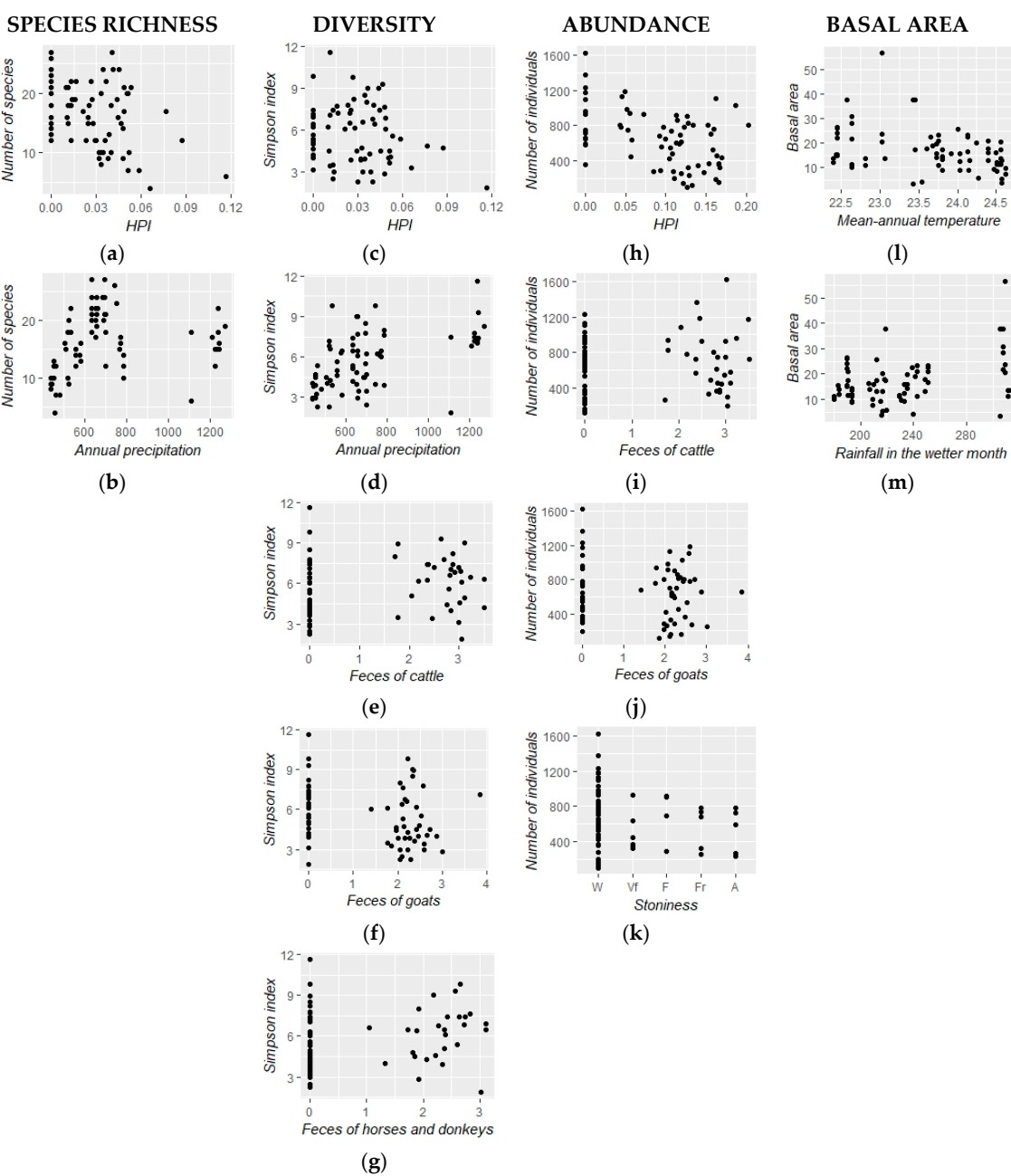

**Figure 5.** Relationships among the predictors that partially contribute to the best models of each assessed variable. In (**a**,**c**,**h**) HPI is log-transformed. In (**k**), the horizontal axis represents the stoniness categories: W = Without; Vf = Very few; F = Few; Fr = Frequent; A = Abundant. In (**e–g**,**i–j**) the feces are expressed in grams log-transformed.

Animals exerted an apparent influence on the diversity index. Among the six models with ∆AIC lower than 2, five models included Goats, HPI, and annual precipitation; three models included

horses and donkeys; two models featured cattle; and one model included soil drainage. HPI exerted a significant negative influence on the diversity index (Table 3), wherein more diversity was observed in forests that were separated from villages than in forests near villages, and more diversity was observed in forests near villages with fewer families than in forests near crowded villages (Figure 5c). Annual precipitation exerted a significant positive influence on species diversity (Table 3), suggesting that wetter places are more diverse than drier ones (Figure 5d).

Additional predictors on the best model did not significantly influence the diversity. Cattle exerted a negative influence (Table 3, Figure 5e), and goats and horses showed a positive influence (Table 3, Figure 5f,g respectively).

One extreme value of goat excrement (Figure 5f) strongly influenced species diversity. After replacing this high value by the average value and re-standardizing the variable, the effect of goat excrement was negative but insignificant (Estimate (E) = $-0.4684$, $p$-value ($p$) = 0.73037). Nevertheless, we found no authoritative reason to remove the extremity because it was not an outlier; rather, the value reflects a common situation in areas near goat paddocks.

Factors such as soil predictors, temperature, rainfall in the wettest months, and altitude scarcely affected the richness and diversity in this part of the forest. Soil drainage presented a positive impact (E = 0.09505, $p$ = 0.8239443) in the second-best diversity model with a high goodness of fit ($\Delta$AIC = 0.16825).

Four models substantially influenced the similarity of the species composition (Table 2). The best model included the HPI and annual precipitation. In this model, both predictors exerted significant negative effects (Table 3), thus, the species composition among the plots was more heterogenous when the HPI increased. The similarity index, which represents the species similarity among plots in the same cluster, must be high among nearby plots; therefore, the greater heterogeneity might be explained by local intervention, which affects only in one plot in the cluster.

The annual rainfall exerted a negative influence on species similarity; thus, the species composition became more heterogenous under wetter conditions.

HPI and stoniness were predictors of all three models, in terms of number of individuals with $\Delta$AIC below 2. Goats, horses, and cows were each presented in two of these models.

Table 2 also indicates the influences of HPI, stoniness, and animals on the number of tree and shrub individuals. The best model included HPI, cattle, goats, and stoniness, all of which exerted a negative influence. However, only the HPI exerted a significant effect (Table 3). The number of individuals decreased with increasing human pressure (Figure 5h). However, a specific negative influence of cattle and goats cannot be confirmed by our data (Figure 5i,j). The extreme value of goats did not affect the results of this model. The influence of stoniness on the number of individuals was also negative but not significant (Table 3, Figure 5k).

The only variable influenced by temperature was the basal area. The best model (among four models with $\Delta$AICs below 2) included the mean annual temperature and the rainfall amount of the wettest month. The mean annual temperature exerted a significant negative influence (Table 3), implying that the basal area decreases with increasing temperature (Figure 5l).

Precipitation in the wettest month exerted an insignificantly positive effect on the basal area (Table 3) in the best model (Figure 5m); nevertheless, a significantly positive effect on the same parameter was shown by the annual rainfall in the second-best model (E = 0.697, $p$ = 0.0004477). Water availability enhances the growth of trees and shrubs. Animal presence exerted no direct effect on the stand density, although horses featured in the fourth model of the basal area. HPI exerted a significantly negative effect on the basal area in the second-best model (E = $-0.8826$, $p$ = 0.0296971). Concordant with the number of individuals, the basal area was reduced because of wood logging.

## 4. Discussion

### 4.1. Floristic Aspects

With 91 species, our richness results are similar to those of other research performed in the study region: Andrade & Jaramillo [50] and Cueva et al. [51] reported 111 species, Aguirre [49] recorded 58 species, and Espinosa et al. [52] recorded 102 species, including trees and shrubs with DBH ≥5 cm. However, in a compilation of different studies and herbarium reviews, Aguirre et al. [21] reported 184 woody species in the deciduous and semi-deciduous formations of the dry forest in the Loja Province, which is double the number of species recorded herein.

Diversities similar to that reported in this research (Simpson reciprocal index from 1.88 to 11.59) were reported by Aguirre [5,49] and Muñoz et al. [53] in the Ecuadorian part of our study area; likewise, similar diversity values were reported in samples located on the Cerros de Amotape Cordillera in the Peruvian area, which was also included in our research, Linares-Palomino [54] reported Simpson values between 2.04 and 10.48.

Regarding structural aspects, species such as *C. trichistandra* (species palatable to animals), and *C. platanifolia*, (characteristic species of dry forest [5,16]), exhibited a non-habitual distribution of diametric classes. The low numbers of individuals in the lower classes might be associated with the browsing of seedlings [19]. *B. graveolens* is also palatable to animals, but the wood of old individuals is in addition used as a flavoring and repellent [55]. In contrast, we found few *H. chrysanthus*, *P. carthagenensis*, and *L. huasango* individuals in the upper classes, suggesting that these valuable species could have been selectively felled until 1978 [19], i.e., before the region was declared closed area, therefore, the latter result may be a consequence of past utilization.

The numbers of individuals and basal area widely differed throughout the forest (Figure 3d,e). In several cases, the less abundant samples occupied the lowest and driest areas; however, this finding cannot be generalized. For instance, one plot with few individuals (located at 255 m a.s.l where the annual precipitation is 457 mm) spanned 108 ind./ha with a basal area of 3.8 $m^2$/ha, whereas another plot with similar characteristics (247 m a.s.l., annual precipitation 467 mm) spanned 911 ind./ha with a basal area of 15.9 $m^2$/ha. Similar situations were relatively common, and were also found in species composition, indicating that factors other than abiotic factors promote these spatial differences. For this reason, pressure predictors and soil characteristics were included in this study.

### 4.2. Diversity, Similarity, and Structure

In our study, HPI most powerfully predicted the richness and abundance of trees and shrubs in the forest, because it negatively influenced four of the five analyzed parameters (Figure 5a,c,h). The basal area of trees and shrubs was less sensitive to the HPI. Espinosa [52] reported the same effect of anthropogenic disturbance on species richness in a dry forest in the Tumbesian region (approximately 100 km east of our study area). Gillespie [56] also found a significant correlation between these factors in dry forests from Central America, and Sagar [57] reported significant differences in species richness, basal area, and abundance among samples with different disturbance levels in India. Thus, our results confirm the findings of other authors [1,3,7], who identified human presence as the primary cause of dry forest degradation.

Agricultural activities could be most dangerous to this area of dry forest. Corn cultivation is common throughout the area during the rainy season, but more so in higher parts to take advantage of the humidity. This activity means a complete change of land use since it eases sowing, cleaning and harvesting tasks. Wood logging is less practiced nowadays; it is allowed for local use, in which case it is a selective extraction. These could be some causes for the high influence of HPI.

Herein, cattle exerted a negative influence on three of our assessed parameters (but exerted no effect on basal area) (Figure 5e,i). In addition to browsing on the vegetation, cattle can trample the soil, preventing the growth of new seedlings. These observations are consistent with Stern et al.'s [58] report that in two protected areas in Costa Rica (Parque Nacional Palo Verde and Reserva Biológica

Lomas Barbudal), both the diversity and structure of the forests have been significantly influenced by cattle grazing. The same conclusion was reported by Gillespie [56] in Central America, wherein he asserted that cattle reduce the capacity of seeds to germinate and that intensive grazing can generate spiny and unpalatable forests. However, as the livestock density in our study has only been estimated from feces, the respective results have to be interpreted with caution.

In contrast to cattle, goats and horses exerted a positive influence on species richness and diversity (Figure 5f,g). This can be explained by the lower food selectivity of these animals and their input of micronutrients to the soil. These micronutrients favor plant growth, as reported by García-Moreno et al. [59] for *Quercus ilex* L. in a Mediterranean open forest. They found significantly higher N and Mg concentrations in the leaves of trees growing in intensively grazed areas, together with elevated inorganic N in the soil. Nevertheless, the negative influence of goats on the number of individuals is explained by the loss of individuals that are eaten or trampled, particularly when seedlings are small [19], which is common in dry forests; this is because seedlings are especially vulnerable to such losses.

Our results revealed a clear but differentiated effect of livestock, indicating a need for more suitable information regarding livestock activities. Detailed counting of animals may improve the results; however, such counting requires much higher efforts on the part of researchers and farmers. Hence, our method and other similar methods should be used and improved upon in future work.

We found that human pressure heterogenized rather than homogenized the forest; at least, human-induced homogeneity was not identifiable in nearby plots. Considering that the driest areas support the fewest species, the extraction of one individual could promote the loss of that species in the plot. This might explain the differences in the numbers of species among nearby samples in our study, and the consequent heterogenization. These results differ from McKinney et al. [11] who stated that homogenization of a forest is the main effect of anthropic perturbation.

We also found that high annual rainfall increased the heterogeneity among the plots. This reflects the higher species richness observed in the wetter areas than the drier areas of our studied ecosystem. Likewise, in all models with a partial contribution of annual precipitation, the predictor positively affected species richness and diversity (Figure 5b,d).

According to Gentry [7] and Gillespie [56], precipitation is a poor predictor of plant diversity in the dry forest of Central America. Furthermore, Espinosa [52] predicted a negative correlation between precipitation and plant diversity. Nevertheless, both Gentry [60] and Clinebell et al. [61] were able to correlate these two factors when a dry–wet gradient existed in the forest. Considering the altitudinal gradient of our study area (200–1100 m a.s.l.) and that the semi-deciduous dry forest lies directly below the low montane dry forest [5,16], we confirmed a transitional formation between the dry and wet forest in our study area. This transition occurs from 1000 m a.s.l in Lozano [16] and from 900 m a.s.l in Aguirre [5].

The negative influence of stoniness on the number of individuals (see Figure 5k) is probably most relevant in the seed germination stage. When seeds meet unfavorable conditions for germinating, their mortality rate should increase, thereby reducing the number of individuals that reach maturity. Unfortunately, we found no previous study that investigated this predictor; hence, a comparison of our results is not possible.

The mean annual temperature only weakly affected the analyzed factors. The exception was the basal area of trees and shrubs, which was significantly negatively influenced by mean temperature (Figure 5l). This trend can be explained by the increased evapotranspiration of plants at high temperatures, which reduces their growth. Contrary to Espinosa [52], we found no relation between this predictor and species richness; however, as the mean annual temperature is strongly correlated with both annual rainfall and altitude (Table 1), we cannot neglect its effect on any of the assessed factors.

The influence of anthropogenic predictors was rarely reported in our reviewed literature. With the exception of Espinosa [52], the methodologies of these studies differed from ours.

For instance, Gillespie et al. [56] assessed the influence of human activities by correlating them with forest parameters; Stern, et al. [58] compared the situations in areas with and without grazing; and García-Moreno et al.'s [59] study focused on one species.

## 5. Conclusions

The composition and structure of the forest in the study area is primarily influenced by abiotic and human pressure factors. The soil characteristics exert weak effects at most.

Similar to other dry forests around the globe, our models confirmed that anthropogenic pressure has already largely modified the diversity and structure of the dry forest in the central part of the Tumbesian region. In fact, this factor affected all studied parameters. Cattle exert the most damaging effects on species richness, diversity, and abundance. Protective measures, such as prohibiting or restricting cattle in largely affected areas and areas of high biological importance, and reducing cattle ownership per family, should be considered.

Goats do not affect the species richness or diversity in the mature forests of our study area, but they reduce the abundance of individuals. Horses and donkeys did not affect any of the analyzed aspects. However, to identify the long-term effects, assessing the influence on the natural regeneration of the forest is necessary.

We emphasize that our results and conclusions about animal effects should be taken with caution, since we consider that feces, while it provides an approximation of animal number, is subject to high inaccuracy, due to decomposition provoked, for instance, by climatic factors.

Among the abiotic factors, annual precipitation was the most important positive predictor of species richness and diversity in the central zone of the Tumbesian region, despite the briefness of the rainy period.

Human pressure has not affected forest heterogeneity among nearby plots in our study area, where environmental conditions are constant and the human pressure is almost identical in different plots. The annual rainfall is the most important predictor of species heterogeneity in this part of the forest.

The diversity and structural parameters were consistent with similar research on this dry forest. However, complementary studies are necessary to clarify the influencers of characteristic species and to clarify whether unusual behaviors result from human or animal interventions, or arise from environmental conditions.

The most important deficiency in this research is probably the lack of information regarding the number of animals throughout the area, and soil characteristics in the Peruvian part of the study area. When these details are known, a better approximation of the true situation can likely be achieved.

**Supplementary Materials:** The following are available online at www.mdpi.com/xxx/s1. Figures S1–S5. Residuals distribution of the best model of each of the assessed variables. Figure S1. Residuals distribution of the best model of richness. Figure S2. Residuals distribution of the best model of diversity. Figure S3. Residuals distribution of the best model of abundance. Figure S4. Residuals distribution of the best model of basal area. Figure S5. Residuals distribution of the best model of species similarity. Table S1. Dataset of assessed variables and, biotic and abiotic predictors of the central part of the Tumbesian region. Values are given for each sample.

**Author Contributions:** Conceptualization: J.C.O., P.H. and C.I.E.; Methodology: J.C.O., P.H., C.I.E., Z.A.M. and E.C.O.; Formal Analysis: J.C.O.; Investigation: J.C.O.; Data Curation: J.C.O. and E.G.; Writing—Original Draft Preparation: J.C.O.; Writing—Review & Editing: J.C.O., C.I.E., C.Q.D., Z.A.M., E.C.O., M.W. and P.H. Visualization: J.C.O. and P.H.; Supervision: M.W.; Funding Acquisition: J.C.O., P.H., C.I.E. and M.W.

**Funding:** This research was funded by the DFG project PAK 824/B3, project PROY_CCN_0030 under Universidad Técnica Particular de Loja, and project PIC-13-ETAPA-005 under Secretaria de Eduación Superior, Ciencia, Técnología e Innovación (SENESCYT). J.C.O. and C.Q.D. were funded via scholarships of SENESCYT. J.C.O. received support from Erzbischöfliches Ordinariat München. In addition, this work was supported by the German Research Foundation (DFG) and the Technical University of Munich (TUM) in the framework of the Open Access Publishing Program.

**Acknowledgments:** We wish to thank Thomas Knoke for his contribution to data analysis, also to Proofreading Service of TUM Graduate School and Elizabeth Gosling for their contribution with the language revision. We are also grateful to Naturaleza y Cultura Internacional, Parque Nacional Cerros de Amotape, and the communities

Malvas, El Cardo, and Piedras Blancas for allowing us to conduct this research on the reserves of La Ceiba, Cazaderos, Cerros de Amotape National Park, Limones, Piedras Blancas and El Cardo. Furthermore, we would to thank Diego González, Jorge Armijos, Geovanny Cango, Angel Gusmán, Johana Gusmán, Israel Medina, Yodán Zapata, Flavio Olaya, and the different communities for their support in data collection.

**Conflicts of Interest:** The authors declare no conflict of interest.

## Appendix A

**Table A1.** Candidate Models (Constructed from Predictor Variables Only).

| ID | Model | ID | Model |
|----|-------|----|-------|
| 1 | ~1 | 44 | ~1 + Goats + Cattle + Equine + HPI + Stoniness |
| 2 | ~1 + Goats | 45 | ~1 + Cattle |
| 3 | ~1 + Goats + Cattle | 46 | ~1 + Cattle + Equine |
| 4 | ~1 + Goats + Equine | 47 | ~1 + Cattle + HPI |
| 5 | ~1 + Goats + HPI | 48 | ~1 + Cattle + Ann.Prec |
| 6 | ~1 + Goats + Temper | 49 | ~1 + Cattle + Mth.Prec |
| 7 | ~1 + Goats + Altitude | 50 | ~1 + Cattle + Stoniness |
| 8 | ~1 + Goats + Ann.Prec | 51 | ~1 + Cattle + Equine + HPI |
| 9 | ~1 + Goats + Mth.Prec | 52 | ~1 + Cattle + Equine + Ann.Prec |
| 10 | ~1 + Goats + Soil.depth | 53 | ~1 + Cattle + Equine + Mth.Prec |
| 11 | ~1 + Goats + Drainage | 54 | ~1 + Cattle + Equine + Stoniness |
| 12 | ~1 + Goats + Stoniness | 55 | ~1 + Cattle + HPI + Ann.Prec |
| 13 | ~1 + Goats + Texture | 56 | ~1 + Cattle + HPI + Stoniness |
| 14 | ~1 + Goats + Cattle + Equine | 57 | ~1 + Cattle + Equine + HPI + Ann.Prec |
| 15 | ~1 + Goats + Cattle + HPI | 58 | ~1 + Cattle + Equine + HPI + Stoniness |
| 16 | ~1 + Goats + Cattle + Ann.Prec | 59 | ~1 + Equine |
| 17 | ~1 + Goats + Cattle + Mth.Prec | 60 | ~1 + Equine + HPI |
| 18 | ~1 + Goats + Cattle + Stoniness | 61 | ~1 + Equine + Ann.Prec |
| 19 | ~1 + Goats + Equine + HPI | 62 | ~1 + Equine + Mth.Prec |
| 20 | ~1 + Goats + Equine + Ann.Prec | 63 | ~1 + Equine + Soil.depth |
| 21 | ~1 + Goats + Equine + Mth.Prec | 64 | ~1 + Equine + Stoniness |
| 22 | ~1 + Goats + Equine + Soil.depth | 65 | ~1 + Equine + Texture |
| 23 | ~1 + Goats + Equine + Stoniness | 66 | ~1 + Equine + HPI + Ann.Prec |
| 24 | ~1 + Goats + Equine + Texture | 67 | ~1 + Equine + HPI + Stoniness |
| 25 | ~1 + Goats + HPI + Ann.Prec | 68 | ~1 + Equine + Mth.Prec + Soil.depth |
| 26 | ~1 + Goats + HPI + Stoniness | 69 | ~1 + HPI |
| 27 | ~1 + Goats + Temper + Mth.Prec | 70 | ~1 + HPI + Ann.Prec |
| 28 | ~1 + Goats + Ann.Prec + Drainage | 71 | ~1 + HPI + Stoniness |
| 29 | ~1 + Goats + Mth.Prec + Soil.depth | 72 | ~1 + Temper |
| 30 | ~1 + Goats + Mth.Prec + Drainage | 73 | ~1 + Temper + Mth.Prec |
| 31 | ~1 + Goats + Soil.depth + Drainage | 74 | ~1 + Altitude |
| 32 | ~1 + Goats + Drainage + Texture | 75 | ~1 + Ann.Prec |
| 33 | ~1 + Goats + Cattle + Equine + HPI | 76 | ~1 + Ann.Prec + Drainage |
| 34 | ~1 + Goats + Cattle + Equine + Ann.Prec | 77 | ~1 + Mth.Prec |
| 35 | ~1 + Goats + Cattle + Equine + Mth.Prec | 78 | ~1 + Mth.Prec + Soil.depth |
| 36 | ~1 + Goats + Cattle + Equine + Stoniness | 79 | ~1 + Mth.Prec + Drainage |
| 37 | ~1 + Goats + Cattle + HPI + Ann.Prec | 80 | ~1 + Mth.Prec + Soil.depth + Drainage |
| 38 | ~1 + Goats + Cattle + HPI + Stoniness | 81 | ~1 + Soil.depth |
| 39 | ~1 + Goats + Equine + HPI + Ann.Prec | 82 | ~1 + Soil.depth + Drainage |
| 40 | ~1 + Goats + Equine + HPI + Stoniness | 83 | ~1 + Drainage |
| 41 | ~1 + Goats + Equine + Mth.Prec + Soil.depth | 84 | ~1 + Drainage + Texture |
| 42 | ~1 + Goats + Mth.Prec + Soil.depth + Drainage | 85 | ~1 + Stoniness |
| 43 | ~1 + Goats + Cattle + Equine + HPI + Ann.Prec | 86 | ~1 + Texture |

**Table A2.** List of Species in the Study Area.

| Family | Species |
|---|---|
| Achatocarpaceae | *Achatocarpus pubescens* C. H. Wright |
| Anacardiaceae | *Loxopterygium huasango* Spruce ex Engl.<br>*Spondias purpurea* L. |
| Annonaceae | *Annona muricata* L. |
| Apocynaceae | *Aspidosperma* sp.<br>*Aspidosperma* sp. 2 |
| Asteraceae | *Fulcaldea laurifolia* (Bonpl.) Poir. |
| Bignoniaceae | *Anemopaegma* sp.<br>*Handroanthus billbergii* (Bureau & K.Schum.) S.O.Grose<br>*Handroanthus chrysanthus* (Jacq.) S.O.Grose<br>*Tecoma stans* (L.) Juss. ex Kunth |
| Bixaceae | *Cochlospermum vitifolium* (Willd.) Spreng. |
| Boraginaceae | *Cordia alliodora* (Ruiz & Pav.) Oken<br>*Cordia lutea* Lam.<br>*Cordia macrantha* Chodat |
| Burseraceae | *Bursera graveolens* (Kunth) Triana & Planch. |
| Cactaceae | *Cereus diffusus* (Britton & Rose) Werderm. |
| Cannabaceae | *Celtis iguanaea* (Jacq.) Sarg.<br>*Celtis loxensis* C.C. Berg |
| Capparaceae | *Colicodendron scabridum* (Kunth) Seem<br>*Cynophalla flexuosa* (L.) J.Presl<br>*Cynophalla* sp. |
| Caricaceae | *Vasconcellea parviflora* A. DC. |
| Celastraceae | *Salacia* sp. |
| Combretaceae | *Terminalia valverdeae* A.H. Gentry |
| Convolvulaceae | *Ipomoea pauciflora* M. Martens & Galeotti<br>*Ipomoea* sp. |
| Erythroxylaceae | *Erythroxylum glaucum* O. E. Schulz |
| Euphorbiaceae | *Croton* sp.<br>*Jatropha curcas* L.<br>Spp. 5 |
| Leguminosae | *Acacia macracantha* Humb. & Bonpl. ex Willd.<br>*Albizia multiflora* (Kunth) Barneby & J.W. Grimes<br>*Bauhinia aculeata* L.<br>*Bauhinia* sp.<br>*Caesalpinia glabrata* Kunth<br>*Caesalpinia spinosa* (Molina) Kuntze<br>*Centrolobium ochroxylum* Rose ex Rudd<br>*Chloroleucon mangense* (Jack.) Britton & Rose<br>*Erythrina velutina* Willd.<br>*Geoffroea spinosa* Jacq.<br>*Leucaena trichodes* (Jacq.) Benth.<br>*Machaerium millei* Standl.<br>*Mimosa acantholoba* (Willd.) Poir.<br>*Mimosa pigra* L.<br>*Mimosa* sp.<br>*Myroxylon balsamum* (L.) Harms.<br>*Piptadenia flava* (Spreng. ex DC.) Benth.<br>*Piscidia carthagenensis* Jacq.<br>*Pithecellobium excelsum* (Kunth) Mart.<br>*Prosopis juliflora* (Sw.) DC.<br>*Senna bicapsularis* (L.) Roxb.<br>*Senna incarnata* (Pav. & Benth.) H.S. Irwin & Barneby<br>*Senna mollissima* (Willd.) H.S. Irwin & Barneby<br>*Senna spectabilis* (DC.) H.S. Irwin & Barneby. |

**Table A2.** *Cont.*

| Family | Species |
|---|---|
| Malvaceae | *Cavanillesia platanifolia* (Bonpl.) Kunth<br>*Ceiba insignis* (Kunth) P.E. Gibbs & Semir<br>*Ceiba trischistandra* (A.Gray) Bakh.<br>*Eriotheca roseorum* (Cuatrec.) A.Robyns<br>*Eriotheca ruizii* (K. Schum.) A. Robyns<br>*Guazuma ulmifolia* Lam. |
| Meliaceae | *Trichilia hirta* L. |
| Moraceae | *Ficus jacobii* Vázq. Avila<br>*Ficus obtusifolia* Kunth.<br>*Maclura tinctoria* (L) Steud. |
| Muntingiaceae | *Muntingia calabura* L. |
| Myrtaceae | *Psidium guajava* L.<br>*Psidium* sp.<br>Spp. 1 |
| Nyctaginaceae | *Bougainvillea peruviana* Bonpl.<br>*Pisonia aculeata* L.<br>*Pisonia floribunda* Hook. F.<br>Spp. 3 |
| Opiliaceae | *Agonandra excelsa* Griseb. |
| Phytolaccaceae | *Gallesia integrifolia* (Spreng.) Harms |
| Polygonaceae | *Coccoloba ruiziana* Lindau<br>*Triplaris cumingiana* Fisch. & C.A.Mey. |
| Rhamnaceae | *Ziziphus thyrsiflora* Benth. |
| Rubiaceae | *Phialanthus* sp.<br>*Randia armata* (Sw.) DC.<br>*Simira ecuadorensis* (Standl.) Steyerm. |
| Salicaceae | *Prockia crucis* P. Browne ex L. |
| Sapindaceae | *Sapindus saponaria* L. |
| Solanaceae | *Acnistus arborescens* (L.) Schltdl |
| Unknown | Spp. 2<br>Spp. 4<br>Spp. 6<br>Spp. 7<br>Spp. 8<br>Spp. 9 |
| Verbenaceae | *Citharexylum poeppigii* Walp |
| 33 | 91 |

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
