# Peer review of "Influence of Anthropogenic Factors on the Diversity and Structure of a Dry Forest in the Central Part of the Tumbesian Region (Ecuador–Perú)"

_forests, doi:10.3390/f10010031_

Round 1

Reviewer 1 Report

The manuscript “Influence of Anthropogenic Factors on the Diversity and Structure of a Dry Forest in the Central Part of Tumbesian Region (Ecuador–Perú)” described the diversity and structural characteristics of the Dry Forest in southern Ecuador and northern Peru, and analyze the effect of anthropogenic and abiotic factors on variables related with forest structure and composition.

The topic is interesting and relevant for understanding the impacts of human activities on structural aspects of dry forest, a poorly and extended ecosystem in South America.

The introduction needs some restructuration. The sampling design and data analysis is suitable and consistent with the objectives. The results and discussion needs some revision to make these sections clearer.

I am not a native English speaker but I consider that the manuscript needs a language revision as some expressions are not proper.

Introduction

I consider it is necessary to reorganize some paragraphs in order to put the broader context regarding to dry forest at the beginning of the introduction and particular characteristic of the Tumbesian region at the end of the introduction.

Lines 64-73. I suggest moving these two paragraphs to the end of the introduction before the objectives.

Methods

Regarding the description of the study area, are all the anthropogenic disturbances acting over all clusters? Are there any relationship between annual rainfall and the type of human activity or between the altitude and the type of human activity?

In relation to data analysis, why did you not model interactions?

Line 133. What means “stratum”?

Results

I suggest explaining the results describing the effects of factor always in the same order. For example, HPI was the most important factor influencing your results, then, when you describe the results for each response variable you should start explaining the effect of HPI. Also, you should change the order in which you refer the figures starting with the factor that you explain first in the text. For example, if you explain the effect of one factor in the text and refer to Figure 5a, the following Figure that you refer in the text should be Figure 5b.

Regarding the selection of the model, I consider you should select one model per response variable, taking into account that the decision of including a predictor variable in the model is not only related with the model fitness but with the question you want to answer. The importance of a predictor in your results is related with the significance of the effect on your response variable not with the model fitness. Then, as an example, domestic animals had no effect on anyone of your response variables in spite of these variables were included in models with better fitness.

Also, I strongly suggest avoiding speculative comments throughout the results and to move these comments to the section “discussion”.

Line 233. I suggest changing “We identified” for “We recorded”.

Line 234. When you say “Among these individuals” are you referring to the 7815 individuals?

Lines 259-264. Consider to delete this part of the paragraph “the latter result may be a consequence of past utilization…. aforementioned activities.” from the section “results”. It is more appropriate for the discussion. Here you are describing the population structure of these species because they could be indicators of some process or actions on the forest. I suggest explaining this in methods for making this clearer for readers.

Lines 275-277. I suggest changing “Overall, these results confirm a high contribution from fixed and random effects in the local variance, but a lesser contribution to the variations in the Simpson indices and basal area.” for “Overall, these results confirm a high contribution from fixed and random factors to the variations in the richness and number of individuals, but a lesser contribution to the variations in the Simpson index and basal area.”

Lines 285-286. You should avoid speculative comments such as “implying that humans reduced the number of species in the forest, possibly by selective logging”, in this section.

Lines 316-317. This sentence is not clear for me. You affirm that at higher values of annual rainfall there is a lower composition similarity (is it the meaning of “a negative influence”?). Then, at higher values of annual rainfall there is a higher heterogeneity, not a higher homogeneity. I suggest rewriting this sentence to make it clearer.

Line 325. Delete “which can be explained by both wood extraction and livestock browsing.”

Lines 329-330. Delete “Incrementing the stoniness might reduce the number of individuals by creating unfavorable growth conditions”.

Lines 334-336. Delete “This trend can be explained by the increased evapotranspiration of plants at high temperatures, which reduces their growth.”

Discussion

I have a few general comments about the discussion. I agree that the HPI was the most important predictor influencing the studies variables. It would be important to discuss what kind of human activity could be more related with the HPI. What are the main activities in the area? Is livestock grazing a common activity in the villages? Is the logging in the area for commercial purposes implying selective logging or is it for providing firewood? I consider it could help to interpret the results.

I strongly suggest being careful with the discussion of cattle effects because there was no effect. There would be many reasons for the absence of an effect, for example, the range of livestock densities covered by samplings could be not enough to detect the effects. Is the amount of feces a good estimator of livestock pressure? It is a common estimator but it can be variable considering the decomposition rate of the site. You should point that results regarding livestock were unclear.

Tables and figures

I suggest putting the figures in the same order in which they appear in the text.

Table 1. Consider remade the table at light of the comments about results.

Figure 5a. Why HPI vary between 0 and 0.6? In methods you say that the variation was between 0 and 0.01367.

Figure 5c and 5h. I consider you could make a log transformation of feces variables to better show the tendency of most data. In order to homogenize the way of showing the result you could apply the log transformation for all the figures showing the effect of feces variables (5c, d, e, h and i). Although in many cases there is not a significant effect, log transformation could help to show no significant tendencies.

Author Response

Dear reviewer

We are too grateful for take part of your time to review our manuscript and for give us yours valuable comments.

All of them have been seriously treat for our team and you will find them in the PDF file attach, where we explain in detail all changes suggested for you.

We hope that document meets with your expectations to be published.

Sincerely,

Jorge Cueva Ortiz.

Reviewer 2 Report

In general, the paper offers a valuable contribution to a better understanding of humans on the dry forest. It also shows an extensive fieldwork assessment and a detailed and appropriate statistical approach. However, it shows several issues in terms of formatting, writing style and edition. I will number the most common issues and then point most of them in a more detailed fashion. Since the importance of the topic is invaluable, I encourage the author to take the proper steps to improve the quality of this paper.

Style

Paragraphs usually start with a topic sentence which is developed further in the body. But in this paper, it is common to find citations at the beginning of the paragraphs. In order to transmit your message more efficiently, start with the subject or topic, and finish with the citations, this way the reader will prioritize the information instead of the authorship. Moreover, MDPI has a very defined style to cite which reduces the need for writing long lists of authors in the body, thus improving the reader experience. Has this paper also been sent to other journals with the “author, year” format in the body of the article? Or does it come from a thesis?

Examples:

Line 55 Paragraph starts with the name of an author

Line 60 The main topic of the paragraph and sentence is biodiversity, not who described it

Line 64 Starts with author names again. The main message is the different formations.

Line 71 idem

Line 103 idem

Line 128 idem

Line 352 In the discussion section is preferred to start with the results arisen from the research, instead of results from other authors. 

Line 380 idem

Line 400 idem

Line 408 idem

Although starting a sentence naming an author is not wrong, the use of this style in this paper is excessive.

Formatting

The difference between figures build using R and Microsoft Excel is too evident. Please use the same font and size for text in all figures. 

Figure 3: Values in the y-axis should be positioned horizontally. Use the las function in R. Please indicate the unit for the basal area (cm2, m2, etc.) per ha. It might be in the text but figures must be self-explanatory.

Figure 4: Units of (b-g) are lacking and labels for x-axis are inconsistent (only (e) has it)

Table 1: Coefficient estimates, standard error, and significance should be included. 

Correlation table in the Appendix should be in the main body of the paper. It shows very important information to better understand the models tested.

Content

Line 71 Citation corresponds to a paper about calcareous grasslands, not dry forests

Study Area section: Please mention dominant, keystone, rare or endangered species. 

Line 122 to 125 Sentence is vague.

Line 129 Please be consistent using the words “formations” or “vegetation type”

Line 131 Define the strata in a shorter way: “…two vegetation types (deciduous and semi-deciduous) and three density levels (high, medium and low)”

Were there any differences between vegetation types?  (richness, diversity, and basal area cover).

Line 145 It reads “We thus inventoried 24 clusters (20 in Ecuador, 4 in Perú) in 72 plots covering a total area of 25.92 ha.” Shouldn’t it say: “We thus inventoried 24 clusters (20 in Ecuador and four in Peru), containing 72 plots, which altogether cover 25.92 ha”?

Line 153 It is implicit in every scientific journal that the correct Latin name for plants should be used. You can cite but you don’t need to mention the website in the body.

Line 254 It reads “External factors” I recommend the use of “biotic” and “abiotic” factors.

Line 170 Reads “basal area”. Should read “basal area of trees”

Line 177 Specify If Annual precipitation (Ann.Prec) was a total (sum) or a mean.

Line 209 It reads “pressure” should be “biotic”

Language

The writing must be improved in terms of the use of the English language. I strongly recommend sending the paper to an English editor. The paper contains cognates and grammatical mistakes and uncommon wording. This will also result in a more compact and straightforward text.

Example: Line 72 When using the words “Effects of “, the word “on” is needed to specify the object receiving the effect. Recommended: Effect of human activities such as … on the diversity and structure of the Ecuadorian dry forest”

Author Response

(The authors gave the same response as above.)

Reviewer 3 Report

The manuscript is well written, the analysis was carefully designed and the results were very well discussed. After addressing minor changes I suggest below, I believe the manuscript will be ready for the publication.

Please see and address my comments below:

1.      Lines 74-78: Cite some references for “intensive discussion between local livestock farmers and conservationists”.

2.      Line 170-171: Unclear how basal area was calculated?  Forest inventory data was used for the calculation… so was any equation used to covert DBH to basal area? Also for HPI calculation, was B the total basal area in each plot?

3.      Line 171-174: the HPI = 0 means no pressure, now give a sense of how large HPI= 0.01367 would be. It seems a very small value. Also I do not understand what the meaning of “circular calculations”. 

4.      Line 191: Explain what the Simpson reciprocal index is and add reference(s).

5.      Line 200: Again explain the Sorensen index and add reference(s).

6.      Lines 203-204: So which one was used, GLMMs or LMMs or both?

7.      Line 219: PQL. Spell this out.

8.      Line 221: Explain why REML was chosen for the analysis of the predictors.

9.      Figure 3b: It is lacking legend. What are a blue line and a grey buffer zone in this figure?  

Author Response

(The authors gave the same response as above.)

Round 2

Reviewer 2 Report

The most recent version shows a great improvement in all aspects in comparison with the first submission. Comments and corrections of the first version of the manuscript were appropriately assessed by the authors. 

I accept the most recent version of this paper with a minor revision.

I only detect a lack of consistency in the number of decimals in the tables. Usually, only two or three decimals are recommended. 

Author Response

Dear reviewer.

Thank you so much for your comments. The changes were made based on your suggestions in the new version.

Again, we are very grateful for your comments, they have been much useful to improve our manuscript.

The cover letter is attached.
